# User Experience in Cosmetics: Perception Analysis Regarding the Use of an Anti-Aging Moisturizer

Louise Idalgo Vasques and Gislaine Ricci Leonardi *

Faculty of Pharmaceutical Sciences, University of Campinas (UNICAMP), 200, Cândido Portinari St.,
"Cidade Universitária Zeferino Vaz", Campinas 13083-871, SP, Brazil
* Correspondence: gislaine.leonardi@fcf.unicamp.br; Tel.: +55-19-35217067

**Abstract:** One of the most important characteristics of aging skin is dehydration, which is why the use of moisturizing products is very important, especially with increasing age. Thus, the user's experience when using a product is interesting for the companies to develop specific cosmetics not only considering the physiological needs of each skin, but also according to the preference of a group if there is any. For this, a moisturizer was developed, containing an antioxidant active, ascorbyl tetraisopalmitate, whose sensory characteristics were evaluated by 33 Brazilian women between 30 and 60 years old. The results showed that the formulation was well accepted by all subjects, regardless of their age group, initial hydration, or the presence of visible signs of skin aging. It is suggested that the presence of the active ingredient in different concentrations caused a different perception of the formula for specific attributes such as the aqueous residue, film formation, and the feelings of oiliness and stickiness to the skin after application. These results suggest that the perception of the sensory characteristics of the product was more related to the nature and proportion of the compounds than to the age of the subjects.

**Keywords:** anti-aging cosmetics; sensory analysis; skin hydration; user experience

## 1. Introduction

Aging is a degenerative and gradual process that can affect an individual's life at many complex levels. With the advance of technology, medicine, and better living conditions, there has been an increase in life expectancy and, consequently, people began to give greater importance to the appearance that they will reach in old age. Therefore, the cosmetic industry has promptly listened and responded to this need, developing new and more efficient products every year to ensure that users can maintain their health and beauty for a long time [1,2].

The structural changes in aged skin are a cumulative result of the combination of the intrinsic and extrinsic factors which comprise chronologic aging, which presents more functional than morphologic changes, and photoaging, which presents structural and physiologic changes due to chronic sun exposure, respectively [3,4].

Some of the most recurrent external attributes of aging are the appearance of wrinkles and expression lines, dehydration of the hair and skin, sagging, the appearance of hyperpigmented areas, and even skin diseases arising from sun exposure during life. These signs of aging, especially on the skin, greatly affect the self-esteem of individuals, which can lead to cases of anxiety, depression, and other psychological disorders. Therefore, maintaining the external appearance throughout life is very important, since it is associated with physical and emotional well-being [2,5].

In recent decades, and especially after the COVID-19 pandemic, there has been an increase in demand for skincare products, especially the so-called "anti-aging" products, to prevent premature aging and to treat the signs of aging in mature skin. This concern is growing even among younger individuals under 25 years old, since the early treatment

of skin aging is recommended and skincare routines for multiple purposes have become popular with social media [6].

For this, it is necessary to know the difference in the needs of each skin, considering the severity of the aging signs, as well as the preferences of each group regarding the characteristics of the cosmetics that will be used for such purposes. It should be noted that for cosmetic formulations to be effective, the definition of good assets and the proof of clinical effectiveness are just two of the important steps to be considered during the development of such formulations. For the well-being of the consumer and for the continuity of cosmetic intervention, the formulation must also have pleasant sensory characteristics and good spreadability on the skin [7,8].

Among the products available for skin, moisturizers are probably the one skincare product that can transform the condition of the skin dramatically [9]. They are complex formulations designed to maintain the water content of the skin between 10 and 30%. Since the skin is always in motion, water is essential to maintain skin plasticity and barrier integrity, and also for the skin to be tactilely perceived as smooth and soft [10].

Moisturizing cosmetics may contain several active substances to aid in product effectiveness [11]. There are two main mechanisms of hydration, the use of occlusive ingredients and humectant ingredients. Occlusives work to prevent water from evaporating into the atmosphere and, thus, condition the skin. They are usually oily substances capable of forming a physical barrier where water cannot pass through. Examples of occlusive agents are hydrocarbon oils and waxes such as petrolatum, mineral oil, paraffin, and squalene; silicones; vegetal or animal fats such as cocoa butter and lanolin; fatty acids; fatty alcohols; polyhydric alcohols; wax esters; vegetable waxes; and others [9,10].

Humectants work to draw water from the viable layers of the skin into the *stratum corneum*. Naturally occurring humectants found in the dermis include glycosaminoglycans such as hyaluronic acid. Some examples of humectant agents are glycerin, honey, sodium lactate, urea, propylene glycol, sorbitol, hyaluronic acid, etc.; and also some vitamins such as A, B5 (panthenol), C, E, and niacinamide [10].

There is also a third mechanism to make the skin smoother, which is the use of emollients that fill the holes in the *stratum corneum* through swelling. Some examples are alcohols such as octyl dodecanol, hexyl decanol, and oleyl alcohol; and esters such as oleyl oleate, octyl stearate, peg-7, glyceryl cocoate, coco caprylate, myristyl myristate, cetearyl isononanoate, isopropyl myristate, etc. [10,12].

Regarding cosmetic active ingredients, vitamin C, or ascorbic acid, is a very common choice when it comes to the prevention and attenuation of aging signs. Vitamin C is well known for its antioxidative potential, whitening properties, and its role in the synthesis and maintenance of collagen, which is an essential structural protein that acts directly on the support, elasticity, and firmness of the skin. Many reports indicate that there is a depletion of vitamin C in aged or photodamaged skin. Moreover, excessive exposure to oxidative stress caused by pollution or UV irradiation can also be associated with low vitamin C levels [13,14].

However, due to the instability of ascorbic acid in aqueous solutions, several stabilization strategies are necessary for incorporating the vitamin into cosmetic formulations, such as maintenance of the formula's acidic pH, protective packaging against oxygen and light exposure, and sometimes packages that are able to separate the aqueous portion from the ascorbic acid powder, mixing them only at the moment of application, which is an effective strategy, but tends to make its production more expensive and, consequently, increases the final price of the product [13,14].

In this sense, this study aimed to evaluate the sensory characteristics of cosmetic formulations containing moisturizing and antioxidant agents in different concentrations in a population of 33 Brazilian women between 30 and 60 years old, who presented or did not present signs of skin aging.

The active ingredient chosen for this study is the esterified Vitamin C derivative molecule, ascorbyl tetraisopalmitate (hereafter called VC-IP®). Different from other deriva-

tive molecules of vitamin C, VC-IP® has lipophilic nature, which means that this molecule is oil-soluble, instead of water-soluble. VC-IP® has been shown to be chemically stable, presents moisturizing effects on the *stratum corneum* and epidermis, and has the capacity to increase the eco-density of the dermis, reduce skin roughness, and improve hyperpigmentation areas when applied to human skin [14,15].

This study is justified by the importance of evaluating the sensory experience of moisturizers for mature skin and the scarcity of scientific studies that evaluate the experience of users with high-fat cosmetic products.

## 2. Methods

For this study, three cosmetic formulations were developed: A, B, and C (Table 1). The formulations consist of oil-in-water (O/W) emulsions of white color, no fragrance, with medium-high viscosity, and medium-high fat composition, which were added with an ascorbic acid derivative, ascorbyl tetraisopalmitate (VC-IP®), at different concentrations: 0 (placebo formula), 5, and 15%. A preliminary stability test was carried out where the formulations were evaluated at different temperatures (room temperature—23 ± 2 °C, refrigerator—5 ± 2 °C, and lab oven—45 ± 2 °C) for 90 days.

**Table 1.** Composition of the developed cosmetic formulations and their respective percentual.

| INCI Name | Commercial Name | A (%) | B (%) | C (%) |
|---|---|---|---|---|
| Aqua | Water | q.s.p. 100 * | q.s.p. 100 * | q.s.p. 100 * |
| C12-C20 acid PEG-8 ester | XALIFIN-15® | 16 | 16 | 16 |
| Ascorbyl tetraisopalmitate | VC-IP® | 0 | 5 | 15 |
| Polypropylene glycol | Propylene glycol U.S.P | 10 | 10 | 10 |
| Glycerin | bidistilled vegetable glycerin U.S.P | 7.5 | 7.5 | 7.5 |
| Phenoxyethanol (and) ethylhexylglycerin | PROTEG SL® | 0.85 | 0.85 | 0.85 |

* q.s.p 100: quantity sufficient for preparation of a 100% of volume.

The stability test is very important to ensure that the formulation remains stable and effective at varied conditions, and to establish a period of validity for the product, considering that it will remain on the shelf for a considerable time before being used.

Regarding the function of each ingredient, the aqueous portion of the formulation consists of water, which is the vehicle of the formula; and polypropylene glycol and glycerin, which are hygroscopic ingredients, have the ability to absorb water, and act as moisturizer and humectant agents protecting the skin from dryness. The oily portion of the formula consists of the C12–C20 acid PEG-8 ester (Xalifin15®), which is an emulsifying wax that rapidly turns into a very viscous emulsion within the correct temperature and mixing conditions; and the active ingredient, VC-IP®, which is an oil-soluble esterified derivative molecule of vitamin C; and phenoxyethanol (and) ethylhexylglycerin (PROTEG SL®) is the preservative.

The formulations remained stable through the stability test of 90 days in different conditions and were prepared again for the intervention. For that, 33 Brazilian women aged between 30 and 60 years old who presented or did not present signs of skin aging were enrolled and signed a free and informed consent form. The study was approved by the Ethics Committee of the University of Campinas (CAAE: 30677820.8.0000.5404).

Before the beginning of the intervention, the initial hydration of the subjects' skin was measured (Courage & Khazaka Corneometer® CM 825), which indicates the hydration level of the superficial layers of the skin (*stratum corneum*) via measurement of skin dielectric properties (unpublished data).

The formulas were randomly distributed to the subjects and the product was applied to the ventral part of the right and left forearms of the subjects for 60 consecutive days, without interruption, once a day. The subjects were instructed not to use any other cosmetic products at the application site and to avoid sun exposure during the study.

After the intervention period, a sensory evaluation questionnaire of the formulation was sent to the subjects, in which the following scores were assigned to be associated with each attribute of the evaluated formulation. This was an exploratory study of observational nature, with a cross-sectional descriptive approach and structured primary data collection. The questionnaire consisted two types of questions: the fist type intended to evaluate the perceived evaluation of the product's features (odor, color, gloss, fluidity, and softness) and the second type intended to evaluate the perception of the skin sensation during rub-in and after absorption of the product (spreadability, aqueous residue, oily residue, stickiness, and film formation), whereas the definitions of each attribute were based and adapted from Vergilio, de Freitas, and da Rocha-Filho (2022) [16] and can be found in Table 2 for better understanding.

**Table 2.** Definitions of the sensory attributes evaluated by the subjects.

| Sensory Analysis | Attribute | Definition |
|---|---|---|
| Olfactory | odor | When the product is directly smelled, it is possible to detect a specific pleasant or non-pleasant odor. When fragrance is added, it is possible to detect or even identify the scent. |
| Visual | color | Under the incidence of light, the product is white or not white. |
| | gloss | Under the incidence of light, the product reflects or does not reflect light. |
| | fluidity | When placing the product between the thumb and forefinger, and in the presence of pressure, the product is or is not adhesive and does not flow or flows easily. There is some or no resistance. |
| | softness | When placing the product between the thumb and forefinger, and in the presence of pressure, the product is soft and comfortable to touch and does not flow easily. |
| Rub-out | spreadability | After 5 to 10 rotations of the finger on the back of the hand, there is some or no resistance between the finger and the skin. |
| 1 min after-feel | aqueous residue | When pinching the skin on the back of the hand between the thumb and forefinger, there is some resistance. The skin does not appear to be oily. |
| | oily residue | When pinching the skin on the back of the hand between the thumb and forefinger, there is no resistance. The skin appears to be oily. |
| | stickiness | Through repeated movements of touching and releasing the finger from the back of the hand, a certain adhesion is felt or not felt between the two surfaces. |
| | film formation | After complete absorbance of the product, it is possible to feel a thin, soft, and silicone-ish layer over the skin. |

For each attribute, the subjects were requested to give a grade from 1 to 5, where 1 = very poor, 2 = poor, 3 = medium, 4 = good, and 5 = excellent.

## 3. Results

In total, the sensory experience of 33 female Brazilian subjects aged between 30 and 60 years old (average age of 46 years old) were registered and evaluated. All 33 questionnaires were considered valid.

Of the 33 subjects, 48.5% were aged between 51 and 60 years old, 30.3% were aged between 30 and 40 years old, and 21.1% were aged between 41 and 50 years old, with an average age of 46.27 years old.

Regarding the initial hydration, values range from 0 to 120 a.u (arbitrary units); however, it is considered that values under 30 a.u. represent very dry skin, values within 30 and 40 a.u. correspond to dry-to-normal skin, and values over 40 a.u. correspond to hydrated skin. From the 33 subjects, 57.6% presented dry-to-normal skin (average of 34.8 a.u.), 21.2% presented very dry skin (average of 25.2 a.u.), and 21.2% presented hydrated skin (average of 44.04 a.u.), respectively. Most of the subjects were not used to applying multiple cosmetic products, especially on the body, and it was the first time that some of the subjects applied a moisturizer for 60 consecutive days. None of the subjects discontinued the product application before the end of the study. All the previous information is summarized in Table 3.

**Table 3.** Characterization of subjects' profile.

| Information | Category | Characteristics |
|---|---|---|
| General | Sex | All subjects are female |
| | | Minimum age: 30 years old |
| | | Maximum age: 60 years old |
| | | Average: 46.27 years old |
| | Age | Standard deviation: 9.11 years |
| | | Median: 50 years old |
| | | Participants older than average: 18 subjects |
| | | Participants younger than average: 15 subjects |
| | | Very dry skin: 21.2% (average of 25.2 a.u.) |
| Skin-related | Initial Hydration | Dry-to-normal skin: 57.6% (average of 34.8 a.u.) |
| | | Hydrated skin: 21.2% (average of 44.04 a.u.) |

When analyzing the responses of all formulations together as one "general product", since they have the same components, it was possible to observe that the formulations were well accepted by all subjects, with the sensory answers "poor" and "very poor" not being associated with any of the features evaluated.

As for the evaluation of formulations A (placebo), B (5% of VC-IP®), and C (15% of VC-IP®) isolated, it is possible to see a few differences. The answers were divided into two categories: the evaluation of the characteristics of the formulas (Figure 1), and the evaluation of the sensory in the skin after application of the formulas (Figure 2).

The answers related to the formula were similar for all subjects; however, a few slight differences can be observed when comparing the formulas A, B, and C. The features "odor" and "gloss" received the same grade in all groups, being unanimous with grade 5; the color of group C received a slightly lower grade.

The attribute "softness" increased slightly in formulas B and C, with a higher percentage of grade 5 in B.

The "fluidity" attribute decreased in a progressional way, with a higher grade attributed to formula A and a lower grade to formula C.

As for the answers related to the skin sensation, there are attributes that can be considered positive, such as the "aqueous residue", "film formation", and good "spreadability", and attributes considered negative such as a "stickiness" and "oily residue" in the skin.

The aqueous residue and film formation were better evaluated in formula B and lower and C, with A being intermediate. The spreadability of formula C also received a lower grade when compared to A and B. The stickiness and oiliness of the formulas were higher in A and higher in C, respectively.

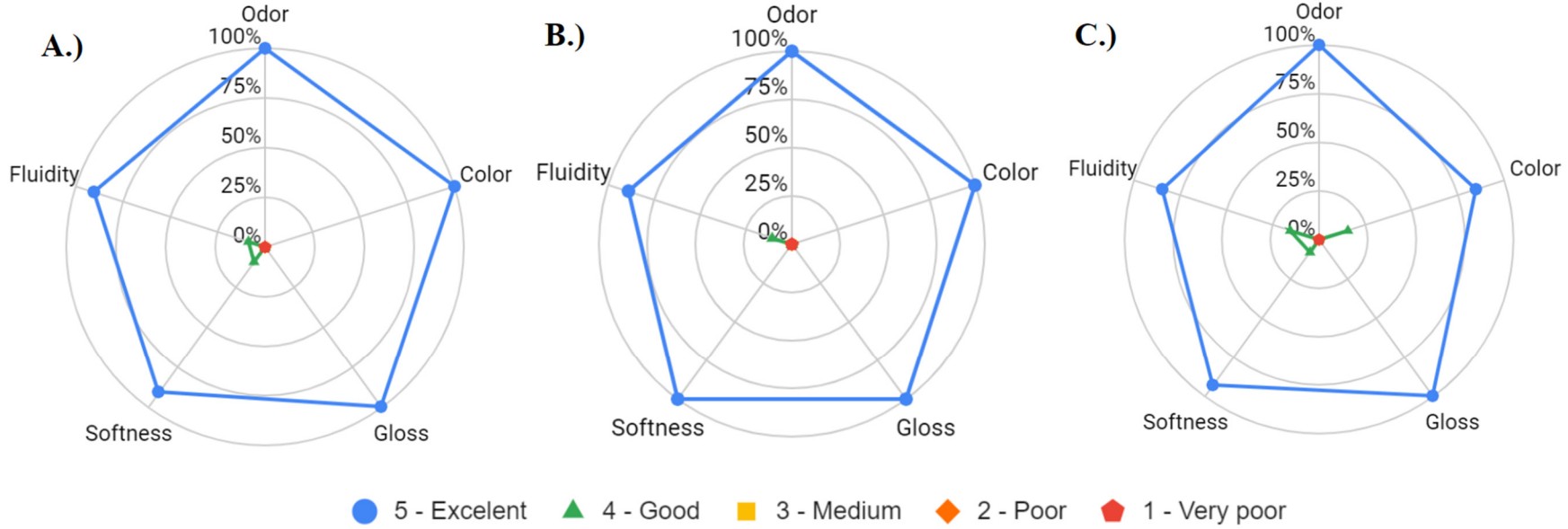

**Figure 1.** Evaluation of the formulations' characteristics. The attributes of the three formulas were evaluated by the 33 subjects who graded them according to their perception. The attributes evaluated were: odor, color, gloss, softness, and fluidity. The possible grades were 5—excellent, 4—good, 3—medium, 2—poor, and 1—very poor. (**A**). Evaluation of formula A, placebo; (**B**). evaluation of formula B, containing 5% of VC-IP®; (**C**). evaluation of formula C, containing 15% of VC-IP®. The more spread out an attribute is in the area of the graphic, the higher the percentage of answers that it received. The grades that do not appear in the graphs were not attributed to any of the characteristics. The percentage is related to the number of subjects, with the total number of subjects—33 women—being 100%. Source: from the authors.

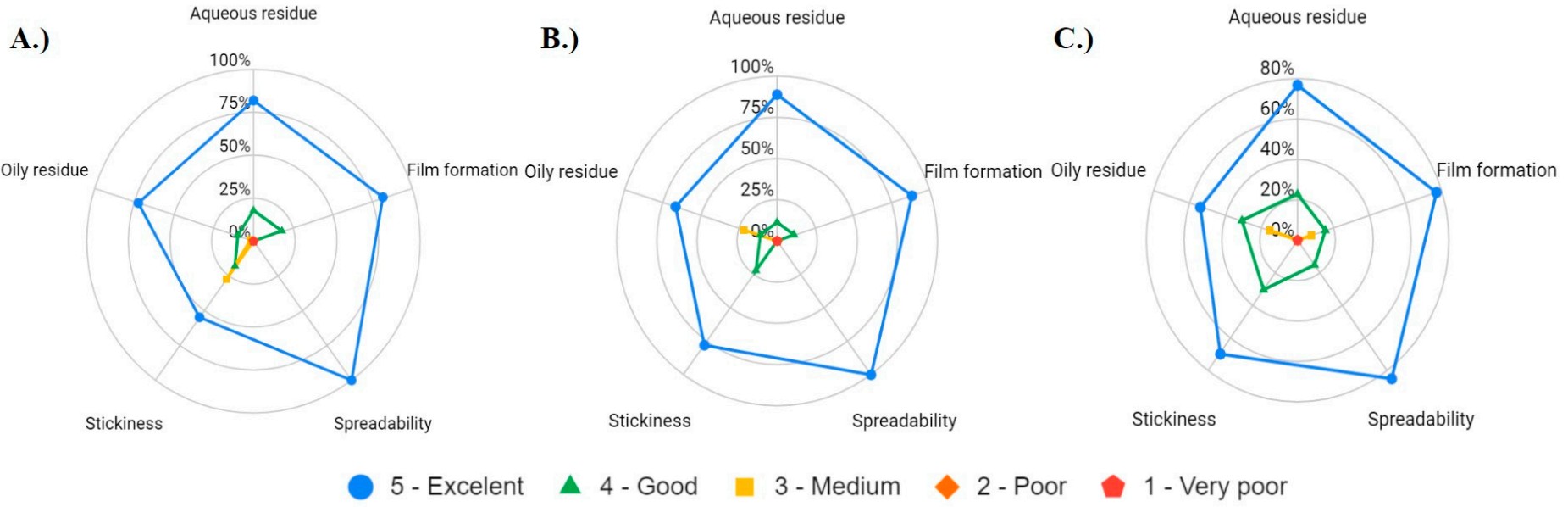

**Figure 2.** Evaluation of the skin feeling after application of the formulas. The attributes of the three formulas were evaluated by the 33 subjects who graded them according to their perception. The attributes evaluated were: aqueous residue, film formation, oily residue, stickiness, and the spreadability of the product. The possible grades were 5—excellent, 4—good, 3—medium, 2—poor, and 1—very poor. (**A**). Evaluation of formula A, placebo; (**B**). evaluation of formula B, containing 5% of VC-IP®; (**C**). evaluation of formula C, containing 15% of VC-IP®. The more spread out an attribute in the area of the graphic is, the higher the percentage of answers that it received. The grades that do not appear in the graphs were not attributed to any of the characteristics. The percentage is related to the number of subjects, with the total number of subjects—33 women—being 100%. Source: from the authors.

## 4. Discussion

### 4.1. Cosmetic Preferences versus Skin Needs

There are many types of cosmetic formulations available on the market today, which can vary according to technology, price, brand appeal, cosmetic form, and audience. This makes it more difficult to choose the right product that can be effective for a particular condition and still perfectly fit a personal preference.

In fact, most consumers do not exactly know if they have a preference or not for a specific type of product, and are most likely to buy products under the influence of the brands' advertisements, TV commercials, social media influencers or celebrities, or dermatologist's recommendations, or even buy products by the label, color, or scent in drug stores and supermarkets.

Another factor is cultural influence. Brazil has currently occupied one of the top positions as a consumer of beauty products in the world, with the most consumed products being fragrances (perfumes), male products, and deodorants (second in world ranking), followed by children's products and sun protection products (third), hair products (fourth), oral hygiene products (fifth) and bath products, make-up (seventh), and depilatories (tenth). Regarding skin products, Brazil remains stable as the eighth largest consumer market [17].

The study conducted by Cho et al. (2017) shows the consumer behavior of 1,015 Korean subjects (with 73.2% being females; mean age, 32.5 years old). The type and number of cosmetic products commonly used by the subjects seem to be higher than the typical Western use. Products such as cleansing oils, toner, essence, serum, day and night moisturizers, nutrition cream, milk lotion, eye cream, mask packs, sunscreens, and make-up are some of many daily used products. Additionally, the education level had an influence on the frequency, amount, and knowledge of the functionality of each product; for example, make-up, sunscreen, and functional cosmetics were significantly correlated with the level of education. Korea is the ninth largest consumer in the world ranking [17,18].

Additionally, with the increase in cosmetic-related allergies, the "green wave", and the cruelty-free movement, it is also possible to note that the cosmetics market is undergoing some transformations in the last decade, with a growth in the demand for products with "clean formulas" containing mostly natural ingredients and of organic origin, without the addition of chemical preservatives, artificial fragrances, and pigments, and for certified brands [19].

Regarding skin needs, the skin was once divided into four types: oily, normal, dry, or combination skin; however, this type of classification is based on subjective descriptions and not objective measurements, which makes them a little vague. Moreover, as we age, the skin undergoes physiological alterations and with that, the needs of the skin also change [20].

For instance, sebum production is a very variable attribute. Different from young skin, which still has a great action of hormones, and which presents more sebum production, mature skin has a low production of sebum and ends up losing more water to the environment when compared to younger skin. Sebum production can also change regionally, since the skin is the biggest organ in the body and, seasonally, increases in hot weather and decreases in cold weather [20–23].

In this way, the production of moisturizing cosmetics with a higher proportion of fat compounds (more occlusive) could be more interesting for mature skin to help keep water in the skin layers and, thus, maintain the volume and hydration of the skin [23].

Based on this premise, moisturizers for younger skin with greater oil production tend to have a lighter consistency and an oil-free formula. However, the personal preference for these types of moisturizing formulations based only on the needs of the skin may not satisfactorily meet the sensation on the skin of individuals, which means that younger individuals will not always necessarily prefer light moisturizers and older individuals will prefer heavier and richer moisturizers.

*4.2. What We Found in This Study*

Regarding the results found in this study, we can sum up the considerations below.

The odor of the formula, here, was actually evaluated by the absence of odor, since no fragrance was added to the product in order to observe if the ingredients would influence the final odor. The same concept was applied to the color, since no pigments were added. The active ingredient (VC-IP®) did not seem to cause any odor or gloss alteration to the formula; however, it may have some influence on the formulation's color, which was not perceived by many subjects.

The increased value in formula C's softness may be related to the addition of the oily nature of the active ingredient, adding an emollient texture to the product. The fluidity can be also related to the addition of the active ingredient, leaving a more fluid texture with the addition of an oil compound in C [24].

The aqueous residue is complex because it depends on the initial condition of the skin, with dryer skin absorbing much more of the product than hydrated skin. However, it is suggested that the composition of the formulas containing an oily ingredient might be more difficult to instantly absorb in the skin, needing a longer time until complete "drying" [23].

The feeling of film formation is the sensation of having a very thin layer of product protecting the skin, almost like a second skin. The results suggest that when the percentage of the active ingredient is higher, that may alter this feeling, but not in a way of causing discomfort to the skin.

The spreadability of a formula depends on the nature of the compounds and the viscosity of the final product, which tends to be more difficult to spread when there is a higher viscosity [24]. Here, we can observe that even though formula C had a more fluid perception, the nature of the ingredients might have made it more difficult to spread the formula. Since all the formulations were emulsions, it has been noticed in other studies that in the case of the formulation of emulsions, oils, butter, and waxes can influence the ease of application of products, with greasy ingredients especially influencing the attributes of spreading, penetrating, oily residue, and stickiness [23]. It was also observed that thickening agents can influence firmness and adhesiveness [8].

Moreover, these "feeling" and "residue" concepts are somehow difficult and abstract to understand, so they can provide only an idea of the sensation felt in the subject's skin. The same thing happens when it comes to understanding the questions and grading the attributes according to personal perspective.

Based on this premise, sensory tests associated with modern tools to capture physiological and emotional responses from user exposure to products have been increasingly used to understand human behavior. Cameras, motion sensors, and computers to detect pose estimation, recognition of patterns, contraction/dilation of the pupil, body and skin temperature, increase of heart rate, and facial expressions, among other behaviors, can sometimes tell more about the user experience or buying intention in a store than their actual answers to satisfaction questionnaires, since the reactions are involuntary and do not require an understanding of the purpose of the study [25].

Here, we want to emphasize that a "lower" grade does not correspond to a bad grade, since all answers attributed to the formulas were classified as excellent, good, and medium, with the attributes "Poor" and "Very poor" not being attributed to any characteristics of all formulas.

The age of the subjects did not seem to be a factor when choosing an answer regarding the formulations, since it was not possible to find a pattern of response when answering a specific attribute and associating it with a certain age range.

In contrast with the expected, the younger subjects (between 30 and 40 years old) did not prove to be unsatisfied with any of the formulas for being too heavy in texture or too rich in greasy compounds.

The initial hydration condition did not seem to interfere in the sensory perception of the subjects as well.

Additionally, none of the 33 subjects presented skin irritation, desquamation, redness, or any other possible adverse reactions from the formulations used. Additionally, none of the subjects discontinued the study.

This sensory questionnaire was applied as a complementary perception study together with other clinical measurements that are not described in this article, and, nevertheless, they do not aim to prove the effectiveness of the formulations, but rather study the perception of the products in the skin as a satisfaction survey.

## 5. Conclusions

Aging is a complex process that can affect individuals on multiple levels (structural, physiological, psychological, sociological, etc.). The motivation and the way people consume cosmetic products can also vary with age, sex, the geographic region, cultural roots and aspects, and educational level, among other factors.

Moisturizing products are one of the most consumed products in the world and their use is even more important with increasing age due to the natural skin dehydration that occurs during one's life. Moreover, moisturizers are probably the one skincare product that can transform the condition of the skin dramatically.

One of the most common active ingredients to improve the moisturizing effects and also help to prevent and attenuate aging signs in the skin is vitamin C, due to its antioxidant and whitening characteristics, as well as its role in the synthesis and maintenance of collagen. However, due to vitamin C instability in aqueous solutions, which usually requires many strategies of stabilization, derivative molecules that are fat/oil-soluble are being tested in order to increase the stability and reduce the product price. Here, ascorbyl tetraisopalmitate (VC-IP) was used as an alternative to ascorbic acid, and it has shown good sensory features.

The sensory characteristics of moisturizing products are very important to the adherence to study, and it is suggested that attributes such as low spreadability, oiliness, and residual stickiness in the skin are considered negative to users of all ages.

The formulas developed in this study fulfilled the sensory attributes evaluated in a satisfactory way. It is suggested that the higher the proportion of greasy and oily ingredients such as the main active compound of the formulas, the more alterations can be observed in the sensory features.

The age factor did not show very distinguished differences among the preferences reported in the questionnaire. It is suggested, therefore, that there was no preference for a formulation with characteristics defined by a group of the same age. The different answers regarding giving a lower grade for an attribute were isolated and did not have similarities among subjects of the same group being associated with personal preferences or personal perceptions.

**Author Contributions:** L.I.V.: conceptualization, methodology, validation, formal analysis, investigation, data curation; writing—original draft, review and editing, visualization. G.R.L.: conceptualization, visualization, supervision, project administration, final approval, and funding acquisition. All authors have read and agreed to the published version of the manuscript.

**Funding:** This work was funded by the São Paulo Research Foundation (FAPESP- 2020/08516-0) and "Coordenação de Aperfeiçoamento de Pessoal de Nível Superior–(CAPES)"–Brazil–Finance Code 0001.

**Institutional Review Board Statement:** The study was conducted in accordance with the Declaration of Helsinki, and approved by the Ethics Committee of the University of Campinas (protocol code CAAE: 30677820.8.0000.5404 approved in July 2020).

**Informed Consent Statement:** Informed consent was obtained from all subjects involved in the study.

**Data Availability Statement:** The data presented in this study are available on request from the corresponding author. The data are not publicly available due to the protection of personal information of research subjects.



**Conflicts of Interest:** There is no conflict of interest.

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
