# Peer review of "User Experience in Cosmetics: Perception Analysis Regarding the Use of an Anti-Aging Moisturizer"

_cosmetics, doi:10.3390/cosmetics10010033_

Round 1

Reviewer 1 Report

Discussions and results are based exclusively on the completion of some forms by the subjects, without more evaluation. The lot is not presented completely...in addition to age, initial evaluations of the skin, s hydration status and skin type had to be carried out. 

Author Response

Dear Editor,

Thank you very much for your review, we have carefully checked your comments and revised the manuscript.

As requested, we have added more detailed information to improve the results of the paper such as the initial mean hydration of the group assessed by the (Courage & Khazaka Corneometer® CM 825) for a better understanding of the group’s skin profile. Those pieces of information can be found in Table 2.

All the other items of the paper: introduction, methods, results, and discussion were supplemented in order to better elucidate the study.

Sincerely,

Louise I. Vasques

Reviewer 2 Report

The subject of the paper is in line with the topic of the journal and relevant to the contents of the article. Due to the conducted experiment, the paper is original, carries significant scientific knowledge, and the results have a practical application and can be used in the industry, for example the cosmetics industry. In particular, the topic is very important for practical applications  (for example in cosmetic formulations for the treatment of hyperpigmentory disorders). The manuscript is technically correct. The introduction outlines the problem and the research objective. The research materials and methods are relevant and well-ordered throughout the paper. The paper clearly outlines the method of collecting research materials. The results are complete, appropriately described and novel.  The tables contain all the necessary information, are appropriately captioned and clear. Units and abbreviations are explicit. The reference materials are well-selected and up to date. The references list contains perhaps (????) 25 items and it is a representative source for the discussed topic. 

In my opinion, the subject matter of the manuscript is appropriate for consideration by the Cosmetics. In general, the topic is interesting, the approach is suitable for a this journal. 

But I have the following comments:

1. I have  objections to the statistical methods. There is no statistical study of the results. There were no statistical analysis given in the paper.

2. Please provide more details on the raw materials used in creams (producer, trade name)

3. The publication does not adheres to the journal’s standards. Perhaps text layout is preserved in accordance with the requirements of the editorial, but the formatting is incorrect in many places. Captions of tables are not made in accordance with the editorial recommendations. The list of references is incorrectly formatted. I suggest you check the formatting of all the text.

Author Response

Dear Editor,

First of all, thank you for the revision and kind comments. We have taken all of them into careful consideration and we are glad that we could get your interest in our paper. 

Regarding your considerations, we understand that a formal statistical analysis will be relevant to improve the quality and confidence of our results, however, this was an exploratory study of observational nature, with a cross-sectional descriptive approach and structured primary data collection, so even though numbers could be attributed to the qualitative responses, they might not project the real quantitative results because the study was not developed around this objective.

Regarding the information on the materials used, we have added a column with the commercial name of all ingredients used in table 1.

Regarding the formatting of the text, we have checked and corrected the places that were in non-conformity with the Journal’s standards.

Sincerely,

Louise I. Vasques

Reviewer 3 Report

This work deals with the sensory analysis of an anti-aging and moisturizing cream for use in cosmetics applications. The evaluation of sensory attributes in cosmetics are important as can positively or adversely influence the attitude of consumers to a product regardless of its possibly very good quality from the cosmetic viewpoint.

Nevertheless, a minor revision should be done.

1.     There is a lack of explanation / definition about the sensory attributes; for example,

“Spreadability: After 5 to 10 rotations of the finger on the back of the hand, there is some or no resistance between the finger and the skin”. I suggest to add this information in the section “Materials and Methods”. It helps to clarify the research methodology.

2.     The attribute “viscosity” is not very clear. In cosmetics, viscosity is a parameter which is evaluated with a rheometer. Could it be “fluidity”, “cohesiveness”, “thickness”? Please, explain this attribute.

3.     Authors are also encouraged to add some up-to-date literature references related to the issue. The inclusion of following literature is recommended:

Moravkova T, Filip P. Relation between sensory analysis and rheology of body lotions. Int J Cosmet Sci. 2016 Dec;38(6):558-566. doi: 10.1111/ics.12319.

Huynh A, Garcia AG, Young LK, Szoboszlai M, Liberatore MW, Baki G. Measurements meet perceptions: rheology-texture-sensory relations when using green, bio-derived emollients in cosmetic emulsions. Int J Cosmet Sci. 2021 Feb;43(1):11-19. doi: 10.1111/ics.12661.

Vergilio MM, de Freitas ACP, da Rocha-Filho PA. Comparative sensory and instrumental analyses and principal components of commercial sunscreens. J Cosmet Dermatol. 2022 Feb;21(2):729-739. doi: 10.1111/jocd.14113.

Author Response

Dear Editor,

Thank you for your revision and comments. We have taken full and careful consideration of all of them. Regarding the items that you have highlighted, we answer each one separately below. 

Regarding the definition of the sensory attributes, we have added Table 2 containing the explanation of each attribute used separated by categories. 

Since we have added Table 2 with the definitions of each attribute, we decided to change the term “viscosity” with the term “fluidity’ and correct the sentences where the term was misused. We embased those definitions on the paper of Vergilio, de Freitas & da Rocha-Filho (2022) which was a collaboration of one of the members of our research group, so we decided to create a pattern of definitions for further studies.

We appreciated the indication of the papers relative to the study and we have supplemented our manuscript with them when the information was suitable.

Sincerely,

Louise I. Vasques

Reviewer 4 Report

I do not find the scientific soundness of this study. It seems like an interesting manuscript but it needs to be presented as a scientific paper from the beginning (eg abstract is missing results). You are also missing the limitation section.

References and figures seem appropriate. The table needs to be improved (+- is not a sign used for SD)

Author Response

Dear Editor, thank you for the revision. 

We understand and appreciate the considerations you have made. However, this study consisted of an observational analysis when performing an efficacy study of the formulation presented in this paper. Yet, we believe that the information found here regarding the behavior and sensory observation of the subjects to the formulation is relevant to be shared with the scientific community.

Additionally, we have done some modifications and supplementation to the text based on all reviewer's suggestions and we believe that the paper is richer and more comprehensive now. 

Sincerely,

Louise I. Vasques

Round 2

Reviewer 1 Report

As a result of the changes you made, the paper can be published in its current form.

Reviewer 4 Report

this manuscript is now acceptable for publication